# Preparation and Performance of Ultra-Fine Polypropylene Antibacterial Fibers via Melt Electrospinning

**DOI:** 10.3390/polym12030606

**Published:** 2020-03-06

**Authors:** Qiu-Sheng Li, Hong-Wei He, Zuo-Ze Fan, Ren-Hai Zhao, Fu-Xing Chen, Rong Zhou, Xin Ning

**Affiliations:** Industrial Research Institute of Nonwovens & Technical Textiles, College of Textiles & Clothing, Qingdao University, Qingdao 266071, China; 2017021401@qdu.edu.cn (Q.-S.L.); 2017021409@qdu.edu.cn (Z.-Z.F.); chinesezrh@126.com (R.-H.Z.); fuxing1991@gmail.com (F.-X.C.)

**Keywords:** melt electrospinning, PP fibers, ZnO, antibacterial fibers

## Abstract

Polypropylene (PP) fibers are employed commonly as the raw material of technical textiles (nonwovens), and the research focuses on fine-denier fibers and their functionalities. In this work, antibacterial PP masterbatches with different dosage (1–5 wt.%) of nano-ZnO particles as the antibacterial agent were prepared via a twin-screw extruder. The as-prepared PP masterbatches were electrospun on a home-made electrospinning device to afford ultra-fine PP fibers. The morphologies of as-spun ultrathin PP fibers with 16 μm of average diameter were observed by SEM. The structure and element distribution were characterized by means of energy-dispersive spectroscopy (EDS) and Fourier-transfer infrared spectroscopy (FTIR), respectively. There was some zinc obviously distributed on the surface when a dosage of ZnO more than 1 wt.% was used, which contributed to the antibacterial activity. The crystallinity of PP fibers was not affected strongly by the dosage of ZnO based on the differential scanning calorimetry (DSC) heating curves, while thermal decomposition improved with the increase in ZnO content, and the mechanical strength decreased predictably with the increase in inorganic ZnO content.

## 1. Introduction

Polypropylene (PP) is a colorless, odorless, non-toxic organic polymer with chemical resistance, electrical insulation, high-strength mechanical properties, and good wear-resistant processing performance. PP and its composites are applied in many fields such as the machinery, automobile, electronics, textile, packaging, agriculture, forestry, fishery, and food industries [1]. With particular regard to technical textiles, following the rapid development of the chemical, environmental protection, energy, and other emerging industries, the development trend of fine-denier PP fibers is being focused, as well as their functionality [2,3,4,5].

In the last few decades, the electrospinning (e-spinning) technique attracted more and more attention because it is a facile method of preparing ultra-fine and functional fibers [6,7,8]. Based on the properties of precursors, the e-spinning strategies can be divided into three types: solution, melt, and novel solventless e-spinning [9]. In the solution e-spinning processes, the polymer solution is used as the precursor. The jet drawn by electrostatic force leaves the Taylor cone and then solidifies quickly with solvent volatilization [10]. Because of the environment concerns of organic solvents and low efficiency, solution e-spinning cannot be easily applied widely and industrialized. Novel types of solventless e-spinning were developed, in which some liquid materials sensitive to light [11], heat [12], or moisture [13] are used as precursors, while the jets are initiated easily under these corresponding conditions to solidify quickly into fibers. However, these methods need more complicated devices to provide steady initiating conditions. Melt e-spinning utilizes polymer melts as precursors, which is close to spunbond, a spinning process for technical textiles [14,15]. The fibers produced by melt e-spinning are much finer than those produced by spunbond. The e-spinning conditions and fibrous morphologies of various melt polymers were previously reviewed [16]. Nayak et al. explored the melt e-spinning conditions of PP [17] and Kadomae et al. explained the relationship between the tactility and diameter of electrospun (e-spun) fibers [18]. Cho et al. investigated the conditions for melt e-spinning of PP and solution e-spinning of PP with dissolution in decalin, with the best results obtained at temperatures higher than 130 °C, and they also compared the e-spun PP ultrathin fibers produced via two different e-spinning methods with average diameters of 9.6 μm and 0.8 μm achieved [19].

Nano-ZnO is a kind of stable inorganic oxide, belonging to the n-type semiconductor family [20]. The electrons on the valence band of ZnO can accept an energy transition from ultraviolet rays, which can provide broad-spectrum ultraviolet protection [21,22], as well as antibacterial properties [23,24], and this material was verified to be safe and effective in the evaluation of sunscreen [25]. With concerns about the safety of heavy metals, such as nano-silver or its ions, more and more researchers began studying the antibacterial properties of nano-ZnO applied as a non-leaching additive [26], such as in cellulose filled with nano-ZnO to prepare an antibacterial lyocell [27], PP or PE doped with nano-ZnO as an antibacterial food packaging film [28], and antibacterial PP nonwovens with the addition of ZnO nanorods [29].

Nano-ZnO powder as an antibacterial additive was applied in some e-spinning precursors of soluble polymers, such as polylactide (PLA) [30], polyurethane (PU) [31], polycaprolactone (PCL) [32], poly (3-hydroxybutyrate) [33], etc. There are scarce reports on melt e-spinning to fabricate antibacterial fibers by adding nano-ZnO. In this research, a blending composite of commercial nano-ZnO in a PP matrix was prepared using a twin-screw machine, and antibacterial masterbatches were produced. A series of ultra-fine PP fibers with different proportions of nano-ZnO were afforded by melt e-spinning, and the morphology, structure, and mechanical and thermal properties of the fibers were analyzed. The antibacterial effect was also evaluated. The obtained fibers have promising application in the field of hygienic textiles.

## 2. Materials and Methods

### 2.1. Materials

PP (ExxonMobil™ 3155E5) was obtained from Shandong SWT New Material Technology Co.,Ltd. (Yantai, China), with a melt flow index of 35 g/10 min, and nano-ZnO was purchased from Boyu High Technique New Material Technology Co., Ltd. (Beijing, China), with an average particle size of 30 nm.

### 2.2. Preparation of PP Antibacterial Masterbatches

The masterbatches were prepared using a twin-screw extruder with a main feeder and a side feeder (16 mm Benchtop Twin-Screw Extrusion Pelletizing Line, Labtech Engineering Co., Ltd., Samutprakarn, Thailand). The pristine PP was charged from the main feeder and nano-ZnO particles were charged from the side feeder, blended in a ratio of 8:2 to prepare the 20 wt.% high-proportion antibacterial masterbatch. The temperature of each heating zone of the twin-screw extruder was set based on Table 1. Then, this masterbatch containing 20% ZnO was diluted by pristine PP in the same twin-screw extruder, thus obtaining PP antibacterial masterbatches with ZnO content of 1 wt.%, 2 wt.%, 3 wt.%, 4 wt.%, and 5 wt.%. The appearances of pristine PP and ZnO, as well as PP with 1 wt.% ZnO, are shown in Appendix A.

### 2.3. Home-Made Melt E-Spinning Device

A melt e-spinning device was built, as illustrated in Figure 1, which consisted of a heater with a controller, a high-voltage power source (HVPS, Tianjin Dongwen high voltage power company, Tianjin, China), a syringe connected with a nozzle and a pipeline of inert CO_2_ gas to prevent oxidation, and a roller receiver connected to a positive electrode.

The prepared PP masterbatches with ZnO content of 0 wt.%, 1 wt.%, 2 wt.%, 3 wt.%, 4 wt.% and 5 wt.% were put into the metal syringe in turn and heated to 210 °C for 20 min while ventilating CO_2_ gas into the metal syringe. This was followed by turning on the receiving roller and HVPS, with the voltage set to 30 kV. The PP melt exited the nozzle, forming a Taylor cone in the high electrostatic field. Then, the melt jet exited the Taylor cone and solidified into a fiber deposited on the receiver. All PP masterbatches were e-spun to give fibers PP-0, PP-1, PP-2, PP-3, PP-4, and PP-5 with ZnO content of 0 wt.%, 1 wt.%, 2 wt.%, 3 wt.%, 4 wt.%, and 5 wt.%, respectively.

### 2.4. Characterization

The morphologies of e-spun fibers were observed by scanning electron microscopy (SEM, TESCAN-VEGA3, Kohoutovice, Czech). The structures and the elemental analysis of e-spun fibers were characterized by a Fourier-transform infrared spectroscope (FT-IR, Nicolet iS10, Thermo Fisher Scientific, Waltham, MA, USA) and an energy-dispersive spectrometer (EDS, SERIAL#: E1856-C2B, Brno, Czech), respectively. Furthermore, their strength was measured eight times for every sample on a FAVIMAT Fiber Test machine (FAVIMAT, TexTechno, Mönchengladbach, Germany), with a tensile rate of 100 mm/min and a tensile length of 10 mm. The adopted stress–strain curve of every sample was close to the average values of all eight tests (see Appendix A). The thermal properties of e-spun fiber were characterized by means of a thermogravimetry analyzer (TGA)/differential scanning calorimeter (DSC) 3 + (TA Q2000, TA Instruments, New Castle, DE, USA), in which the temperature of differential scanning calorimetry (DSC) was raised from 35 °C to 250 °C under an atmosphere of nitrogen. Every sample was tested three times on the DSC for averaging the thermal enthalpy. The thermogravimetric analyzer (TGA) was also implemented in an N_2_ atmosphere. The temperature was raised from 35 °C to 700 °C, and the heating rate was 10 °C/min. The antibacterial property of the fibers was determined via an agar plate diffusion test based on the standard of GB/T 20944.1-2007 (China), and the inhibitory effects on *Escherichia coli* (E.C.) and *Staphylococcus aureus* (S.A.) were evaluated.

## 3. Results and Discussion

### 3.1. Morphologies of PPS E-Spun Fibers

As shown in Figure 2 and Figure 3, the morphologies of e-spun PP fibers containing different contents of ZnO were observed by SEM. All e-spun fibers with or without ZnO were about 16 μm on average in diameter, which indicated that the e-spinning process and as-spun fibers were not affected strongly when adding inorganic nano-ZnO with content lower than 5 wt.%

SEM with energy-dispersive spectroscopy (SEM–EDS) was employed to show the element distribution on the surface of e-spun fibers. Although there were some ZnO particles embedded inside the e-spun PP fibers, others were distributed on the surface, as demonstrated by the EDS patterns (Figure 4). The content or density of zinc especially increased when the dosage increased. As shown in the FTIR spectra (Figure 5), the peaks at 2946 cm^−1^ and 2862 cm^−1^ were assigned to asymmetric and symmetric stretching vibrations of CH_3_ of PP, and those at 2912 cm^−1^ and 2833 cm^−1^ were assigned to CH_2_ of PP. The peak of 1459 cm^−1^ was attributed to the bending vibration of CH_2_ of PP, and that at 1373 cm^−1^ was assigned to the deformation vibration of CH_3_. The peak at 3435 cm^−1^ of ZnO may have originated from the stretching vibration of residual OH or moisture, as, after blending with PP in the extruder, it disappeared because of the high temperature or low dosage. All FTIR spectra of the e-spun PP fibers demonstrated that the PP structure, including crystalline type, was not changed [34].

### 3.2. Thermal Properties of E-Spun PP Fiber

Figure 6a shows the DSC analysis of the fiber. The melting temperature of the PP fibers with different content of ZnO was between 168 and 170 °C, which shows that the addition of nano-ZnO particles did not affect the melting point of the PP fibers. The crystallinity of e-spun fibers was calculated as follows:
Χc = ∆H_1_/∆H_2_,(1)
where Χc is the crystallinity, ∆H_1_ is the thermal enthalpy of the sample (J/g) given by measuring the peak area in the thermogram, and ΔH_2_ is the thermal enthalpy of 100% crystalline PP (209 J/g) [35].

As shown in Figure 6b, the crystallinity of the e-spun fibers did not change greatly when the dosage of ZnO was lower than 4 wt.%. However, it decreased remarkably with the addition of ZnO higher than 5 wt.%, which indicated that more inorganic nanoparticles affected the crystallization.

When ZnO nanoparticles were added to PP, the thermal decomposition temperature of the fibers increased. Based on the TG and DTG curves in Figure 7, the rate of fiber decomposition gradually increased with the increase in nano-ZnO content, and the thermal decomposition temperature constantly increased from 438 °C (PP-0) to 461 °C (PP-5). Because nano-ZnO has a large specific surface area, when it is dispersed in the PP matrix, it would inhibit the release of volatile thermal decomposition products, thus playing an important role in forming a barrier layer and further inhibiting the decomposition of the matrix [36].

### 3.3. Mechanical Tensile Properties of E-Spun PP Fibers

Due to their compatibility or dispersion with inorganic compounds, organic polymers, including PP, commonly suffer from a deterioration of mechanical performance when preparing composites or functional materials with inorganic materials. In this work, the introduction of nano-ZnO particles as an antibacterial additive also caused a reduction in the mechanical properties. The stress–strain behavior of the e-spun PP fibers was compared (Figure 8a), and the elongation at break and breaking strength were found to reduce with an increase in the content of ZnO (Figure 8b).

### 3.4. Antibacterial Properties of E-Spun Fibers

The antibacterial mechanisms of ZnO were disclosed under different conditions. Under dark conditions, the antimicrobial activity of ZnO nanoparticles results from the attachment of ZnO to bacterial cell walls and a subsequent release of Zn^2+^ ions into the bacterial cytoplasm [37]. It is generally believed that the electrons (e^−^) on the valence band of ZnO are excited and transition to the conduction band when they are irradiated by light with a photon energy larger than the band gap width, leaving a positively charged hole (h^+^) on the valence band. The e− and h+ react with oxygen, hydroxyl, and water adsorbed on the surface of the substrate materials to form OH·, O_2_^−^, and H_2_O_2_. Among them, h^+^ and OH· have very strong oxidation activity, which can break the chemical bond of most organic materials. Thus, they can decompose various components of microorganisms and be used to kill germs. In addition, O_2_^−^ has a high reduction capacity and also plays a role in antibacterial performance [38,39,40].
ZnO + hv → e^−^ + h^+^(2)
h^+^ + H_2_O → OH· + H^+^(3)
e^−^ + O_2_ → O_2_^−^(4)
O_2_^−^ + 2 H^+^ → H_2_O(5)

The as-spun PP fibers were cut into pieces and put into agar solution (15 mL) which was added to the bacterial solution (the bacterial colony concentration was 1 × 10^8^ colony-forming units (CFU)/mL), based on the national standard of agar plate diffusion (GB/T 20944.1-2007, China). After culturing for 24 h under a constant temperature of 36.5 °C, the inhibition effect of the fiber on *Escherichia coli* and *Staphylococcus aureus* was as shown in Figure 9 and Figure 10. The pure PP fibers were used as control samples (Figure 9a and Figure 10a). There were still many colonies, and there was no antibacterial ring, i.e., no antibacterial effect. The other PP fibers with different proportions of ZnO nanoparticles had obvious inhibition zones, i.e., an antibacterial effect, which was attributed to the nano ZnO particles migrating to the surface of the fibers.

## 4. Conclusions

PP masterbatches containing different contents of nano-ZnO particles were prepared using a twin-screw extruder, and then applied as a melt e-spinning precursor to obtain ultra-fine fibers with antibacterial activity. The e-spun PP fibers were afforded with 16 μm of average diameter, and their structures and morphologies with or without nano ZnO were not changed remarkably. When ZnO dosage was increased, the strength of the fibers decreased. Although the melting temperature and crystallinity of the e-spun fiber were not changed greatly, the thermal stability was effectively improved, as the decomposition temperature increased from 438 °C (PP-0) to 461 °C (PP-5). All e-spun PP fibers with ZnO dosages of 1 wt.%, 2 wt.%, 3 wt.%, 4 wt.%, and 5 wt.% were evaluated in terms of antibacterial performance, where even the 1 wt.% dosage of nano-ZnO particles could grant the e-spun PP fibers antibacterial activity against *Escherichia coli* and *Staphylococcus aureus*.

## Figures and Tables

**Figure 1 polymers-12-00606-f001:**
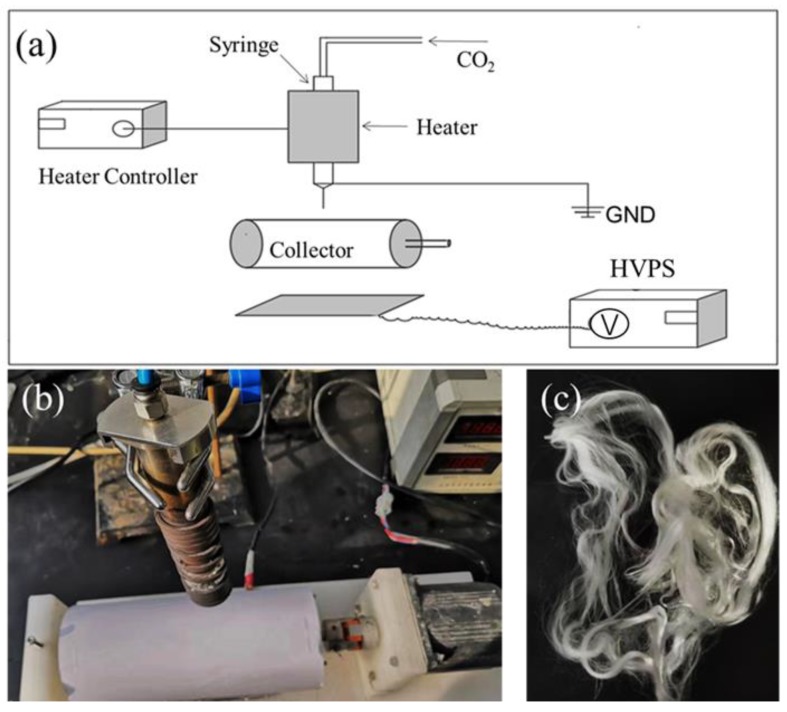
(**a**) Home-made melt e-spinning device; (**b**) heating device; (**c**) melt e-spun polypropylene (PP) fibers.

**Figure 2 polymers-12-00606-f002:**
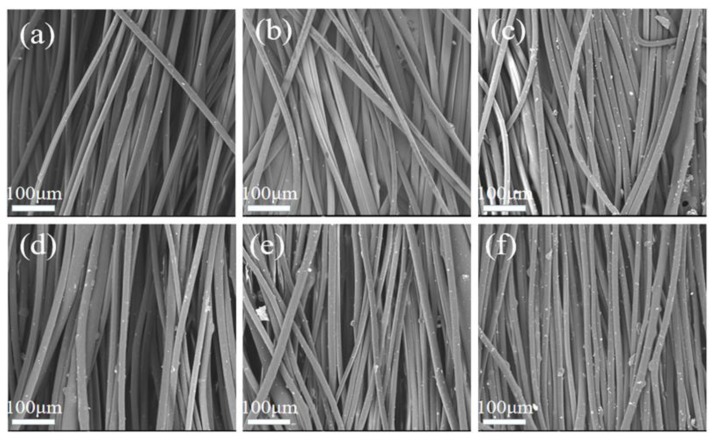
The SEM images of e-spun PP fibers containing ZnO: (**a**) 0 wt.%; (**b**) 1 wt.%; (**c**) 2 wt.%; (**d**) 3 wt.%; (**e**) 4 wt.%; (**f**) 5 wt.%.

**Figure 3 polymers-12-00606-f003:**
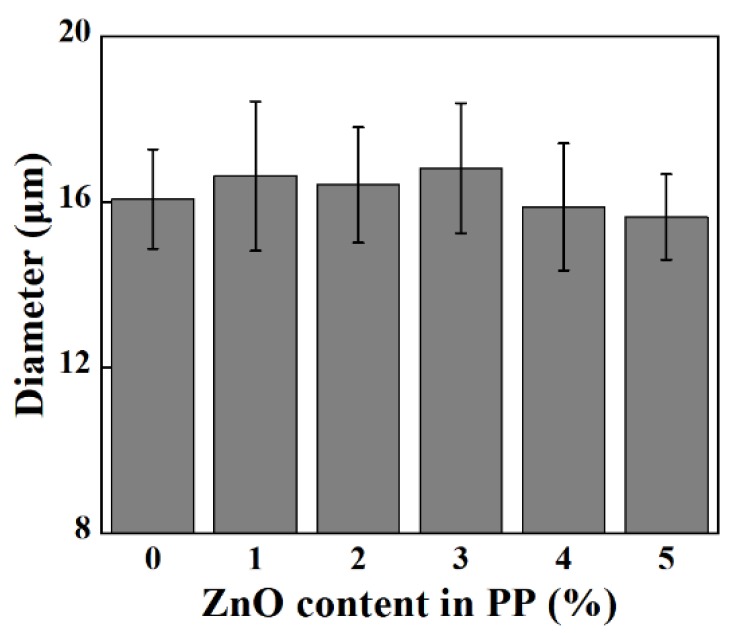
Average diameter of e-spun PP fibers.

**Figure 4 polymers-12-00606-f004:**
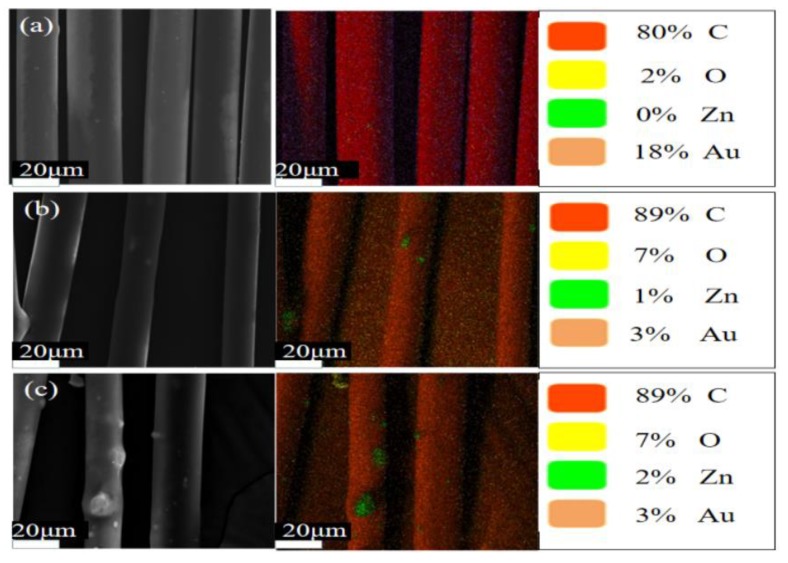
SEM/energy-dispersive spectroscopy (EDS) patterns of PP e-spun fibers containing ZnO: (**a**) 0 wt.%; (**b**) 1 wt.%; (**c**) 3 wt.%.

**Figure 5 polymers-12-00606-f005:**
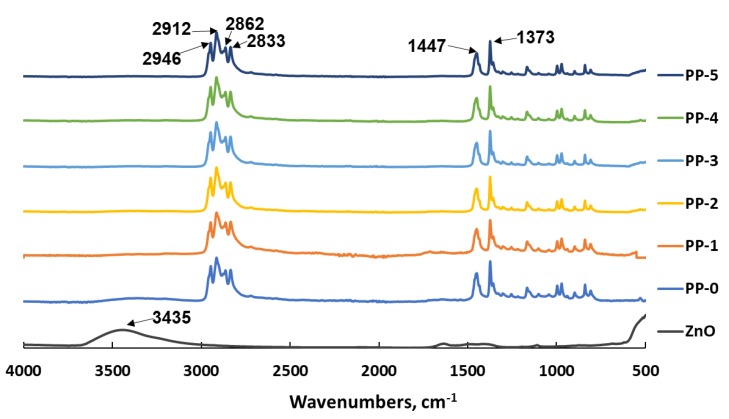
Fourier-transform infrared (FTIR) spectra of ZnO and e-spun PP fibers with different content of ZnO.

**Figure 6 polymers-12-00606-f006:**
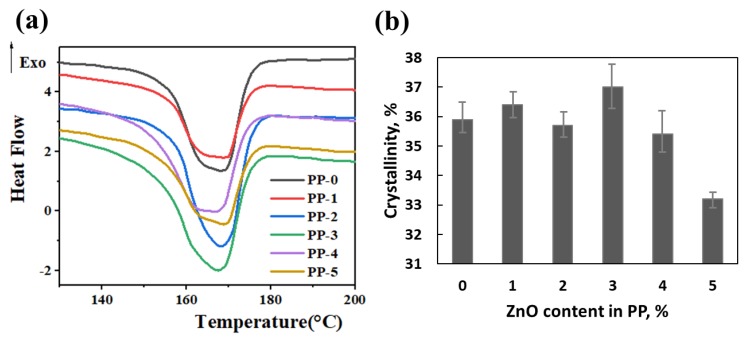
(**a**) Differential scanning calorimetry (DSC) heating curves and (**b**) crystallinities of PP fibers.

**Figure 7 polymers-12-00606-f007:**
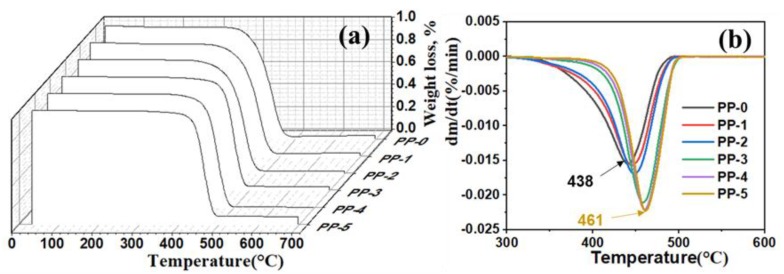
(**a**) Thermogravimetry (TG) traces of as-spun fibers and (**b**) differential TG (DTG) spectra.

**Figure 8 polymers-12-00606-f008:**
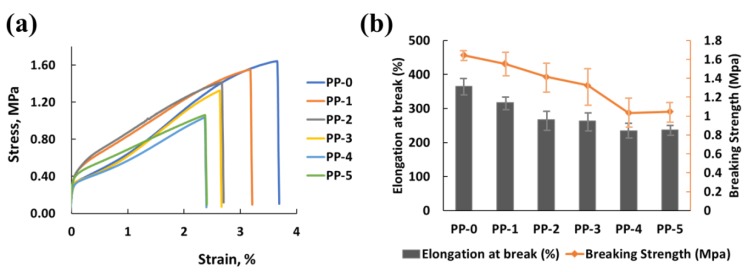
(**a**) Stress–strain behavior, and (**b**) trend of elongation at break and breaking strength of e-spun PP fibers.

**Figure 9 polymers-12-00606-f009:**
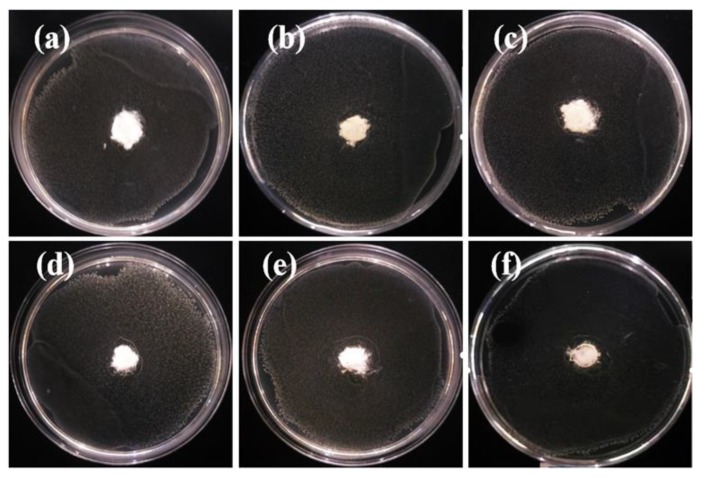
Antibacterial (*Escherichia coli*) activity measurements of e-spun PP fibers containing ZnO: (**a**) 0 wt.%; (**b**) 1 wt.%; (**c**) 2 wt.%; (**d**) 3 wt.%; (**e**) 4 wt.%; (**f**) 5 wt.%. The diameter of all containers was 9 cm.

**Figure 10 polymers-12-00606-f010:**
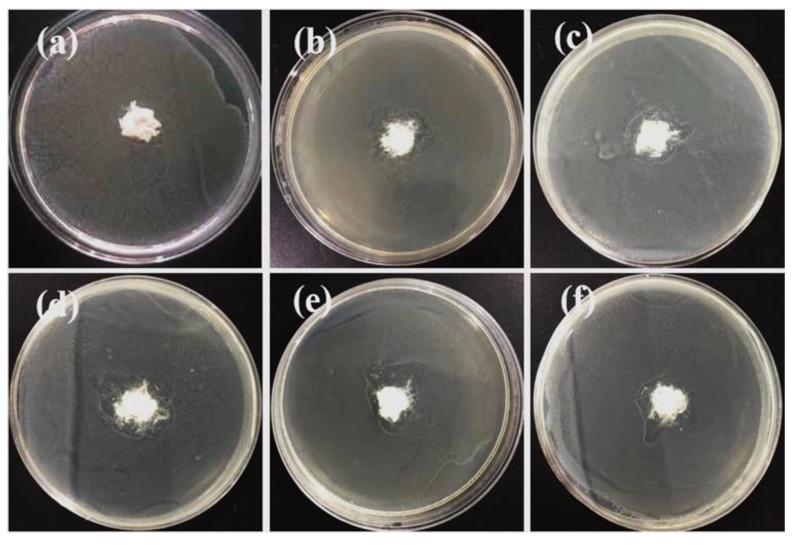
Antibacterial (*Staphylococcus aureus*) activity measurements of e-spun PP fibers containing ZnO: (**a**) 0 wt.%; (**b**) 1 wt.%; (**c**) 2 wt.%; (**d**) 3 wt.%; (**e**) 4 wt.%; (**f**) 5 wt.%. The diameter of all containers was 9 cm.

**Table 1 polymers-12-00606-t001:** The temperatures of heating zones of twin-screw extruder.

Heating Zones	1	2	3	4	5	6	7	8	9	10
Temperature (°C)	170	180	190	200	200	210	210	210	210	210

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
