# Peer review of "Preparation and Performance of Ultra-Fine Polypropylene Antibacterial Fibers via Melt Electrospinning"

_polymers, 2020, doi:10.3390/polym12030606_

Round 1

Reviewer 1 Report

The authors present an interesting article entitled "Preparation and performances of ultra-fine polypropylene antibacterial fibers via melt electrospinning"

The abstract is clear and concise.
The introduction is clear and concise.
The experimental section is clear.
The results and discussion section is ok.
The conclusion is clear and concise.
The references are balanced.

Please change "In solution e-spinning process," to read "In solution e-spinning processes,"
Please change "in nitrogen atmosphere" to read "under an atmosphere of nitrogen"
For chemical formulae numbers should be subscript, please correct "measured in N2."
Please change "Figure.2-3," to read "Figure 2 and Figure 3,"
Please change "The FT-IR spectra of e-spun PP" to read "The FT-IR spectra (Figure 5) of e-spun PP"

3.1.
Please increase the discussionm of SEM-EDS data.
Please increase the discussion of FTIR data (e.g. assign the peaks you observe).

3.2.
When discussing DSC data the authors state "which indicated that more inorganic nanoparticles affect the crystallization" however, the data needs to be collected from multiple experiments to make such an assertion. Please ensure at least 3 repeats of the DSC experiments and include standard deviation error bars in Figure 6b.

3.3.
When discussing mechanical testing data the authors state "In this work, the introduction of nano ZnO particles as
antibacterial additive caused reduction of mechanical properties as well. Compared with stress-strain behavior of the e-spun PP fibers (Figure.8a), and the elongation at break and breaking strength were reduced with content of ZnO increasing (Figure.8b)." however, the data needs to be collected from multiple experiments to make such an assertion. Please ensure at least 6-10 repeats of the mechanical testing experiments and include standard error of the mean error bars in Figure 8b.

3.4.
"h+" should read "H+" in the text and equations.
Equations should be numbered in MDPI format.
Figure 9 and 10 need a note of the diameter of the containers in the figure legends.

Figure 3: please make the blue bars grey/black. Y-axis legend: please insert a space between "Diameter(micrometers)" to read "Diameter (micrometers)"

Author Response

Dear Editors and Reviewers:

Thanks a lot for your letter and comments on our manuscript entitled “Preparation and performances of ultra-fine polypropylene antibacterial fibers via melt electrospinning”. These valuable comments are very helpful for revising and improving our manuscript. We have revised the manuscript carefully in accordance with the Reviewers’ comments, which can be checked conveniently by using “tracked changes” function of Microsoft Word. And some revised parts are highlighted. The point-by-point responses to all comments are listed as follows.

The authors, once again, appreciate for the Editors/Reviewer’s work earnestly, and hope that the responses and revisions could meet with your approval.

Thanks a lot for your positive consideration.

Sincerely yours

Qiu-Sheng Li,

Qingdao University

No. 308 Ningxia Road,

Qingdao 266071, Shandong Province, P. R. China

Tel: +86-138-6390-0519

Reviewer 2 Report

1) English should be revised. A more in-depth check of verbs should be done throught the text, expecially in the abstract section.

2) What about the distribution of ZnO particles over fibers? Is it homogeneous? ZnO particles are embedded inside the structure of the fibers, thus particles surface is no active. How is possible the detection of bacteria? 

3) Antibacterial behaviour of ZnO-fibers is not clear. Authors should clearify this point and explain the mechanism of detection. 

4) Results are merely reported and not discussed in depth. Authors should improve discussion and correlation of results.

Author Response

(The authors gave the same response as above.)

Round 2

Reviewer 1 Report

The authors present an interesting article entitled "Preparation and performances of ultra-fine polypropylene antibacterial fibers via melt electrospinning"

The abstract is clear and concise.
The introduction is clear and concise.
The experimental section is clear.
The results and discussion section is ok.
The conclusion is clear and concise.
The references are balanced.

The authors have extensively revised the manuscript in line with the reviewer's comments and it is thereby improved.

Reviewer 2 Report

Authors provided satisfactory answers. The paper was improved and now can  be accepted in the present form.